# Research Review on Multi-Port Energy Routers Adapted to Renewable Energy Access

**Jinghua Zhou * and Jiangbo Wang**

School of Energy Storage Science and Engineering, North China University of Technology; Beijing Laboratory of New Energy Storage Technology, Beijing Municipal Education Commission, Beijing 100144, China; zh@mail.ncut.edu.cn
* Correspondence: zjh@ncut.edu.cn

**Abstract:** With the continuous development of renewable energy technologies, both domestically and internationally, the focus of energy research has gradually shifted towards renewable energy directions such as distributed photovoltaics and wind power. The penetration rate of renewable energy generation is constantly increasing, at the same time, the elements in the grid are becoming increasingly complex, and large-scale energy storage, as well as a variety of electricity loads such as electric vehicle charging piles and data centers are gradually appearing. Therefore, traditional distribution methods of the power grid are difficult to ensure the stable operation of the power system and cannot achieve efficient integration of renewable energy. Consequently, some scholars have proposed the concept of an energy internet. Compared to traditional power grids, the energy internet employs more comprehensive power electronics and communication technologies, enabling the interconnection of various new and traditional energy sources, and effectively integrating renewable energy. As the core device in the energy internet, the energy router plays a role in energy transformation and distribution, facilitating multi-information interconnection and multi-energy exchange within the energy internet. At the level of distribution network, the energy router can realize the efficient access of various forms of energy and the flexible control and management, which is of great significance for the optimal operation of distribution network. Against this backdrop, this paper reviews the development and current research status of energy routers, systematically analyzes the typical topologies and related control technologies of multi-port energy routers and summarizes and forecasts key issues and future development trends, aiming to provide thoughts and reference for subsequent related research.

**Keywords:** energy internet; distribution network; distributed generation; multiport converter; power electronic technology; smart grid

## 1. Introduction

With the increasingly prominent energy problems and the increasing environmental pollution, wind, light and other renewable energy generation and energy storage are connected to the power grid on a large scale, which alleviates the pressure of the power grid to a certain extent, but because of the intermittent, random and other characteristics of renewable energy, it has an impact on the smooth operation of the traditional power grid, but also makes it difficult to consume renewable energy efficiently. At the same time, a large number of new loads, represented by data centers and electric vehicle charging piles, are connected to the distribution network, which increases the uncertainty and complexity of the operation state of the distribution network, and puts higher requirements on the energy management of the distribution network. It is necessary to optimize the operation scheduling technology of the distribution network, improve the power balance ability and flexible operation level. Therefore, traditional distribution networks require new technologies and equipment to cope with the high percentage of renewable energy

penetration, as well as a large number of new load access brought about by security, stability and other challenges [1–4]. In this context, the concept of energy internet came into being. Energy internet is a large-scale energy network that integrates a large number of distributed generation (DG) and energy storage devices based on the existing power grid. It combines advanced power electronics technology with information technology to realize information sharing, cascade utilization and coordination of energy [5,6], and provides a variety of plug and play interfaces for DG equipment, energy storage equipment and various new types of loads. Thus, multi-directional energy flow can be realized to meet the requirements of the distribution network for the control of the diversity and complexity of electric energy [7,8], which will play a very important role in the construction and development of the future smart grid.

As the core device of energy internet, the energy router is a new type of intelligent power electronic device that combines power electronic conversion technology and information technology [9,10]. The national standard GB/T 40097–2021 "Functional Specifications and Technical Requirements for Energy routers" defines energy routers as with electric energy as the main control object, it has three or more power ports and functions of power conversion, transmission and routing between electric energy with different electrical parameters, which can realize the integration of the electrical physical system and information system, coordinate with the upper system, and control and manage the power supply, energy storage and load accessed by it [11]. The basic architecture of the energy router is shown in Figure 1, which mainly includes the information layer and the electrical physical layer.

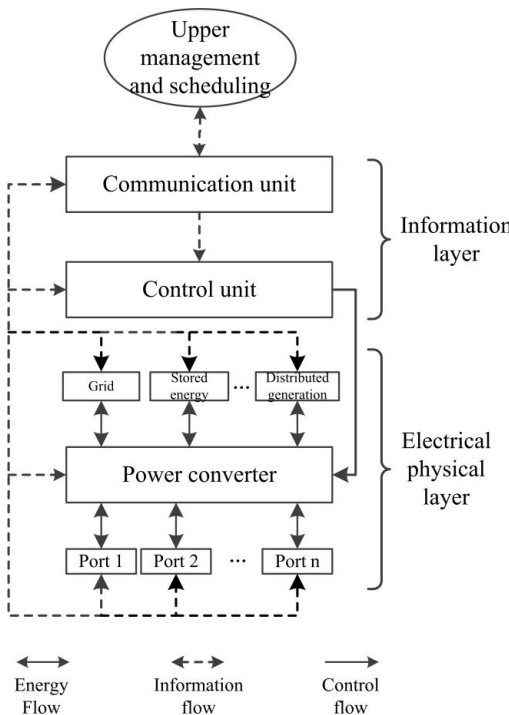

**Figure 1.** Basic structure of energy router.

The information layer is the communication unit and the control unit. Communication unit includes the router internal communication interface and external communication interface in two parts; the internal communication is responsible for the information communication between the combined unit and the controller, and the external communication interface through a variety of forms of communication bus with the external system (other level or superior node equipment) for information exchange; the control unit receives the information from the communication unit and the feedback from the power conversion

unit, and is controlled by the internal power scheduling management unit to realize the multi-direction energy flow, coordination and complementarity within the energy router.

The electrical physical layer is the power conversion unit, which is the hardware circuit part in the general sense, and the power conversion unit is also the core of the energy router, which mainly realizes the power form/voltage level conversion, electrical isolation, and provides multiple plug and play access ports [12,13]. Therefore, as an open energy carrier, the energy router can provide flexible and standardized power electronic interfaces for various distributed energy sources, loads and even power grids. On the other hand, it can collect and control the electricity volume of each port in real time to meet the dispatching requirements of the power grid and provide data guarantee for the effective and stable operation of the energy internet [14], which is an indispensable link in the construction process of the energy internet.

In the literature [15,16], Energy routers are divided into three types according to topology and implementation methods: energy router based on Solid State Transformer (SST), multi-port energy router and Power Line Communication (PLC).

1. Energy router based on SST

The energy router based on Solid State Transformer is mainly used to realize the conversion and control of energy flow among various power forms and voltage levels in regional medium and low voltage distribution networks or microgrids [17], the core of which is SST, also known as power electronic transformer (PET). Compared with traditional transformers, SST integrates electrical isolation, voltage conversion, reactive power compensation and other functions, which can be connected to the AC power grid or directly to the DC power supply, so as to facilitate the grid connection of renewable energy such as photovoltaic power generation [18–20]. Typical SST consists of a high voltage stage, isolation stage and low voltage stage. The high voltage stage converts the primary input voltage into high frequency AC square wave, then transforms the voltage level through the high frequency transformer, and then obtains the target voltage through the low voltage stage for rectification transformation. The typical architecture is shown in Figure 2. Because it has a variety of AC and DC ports and a high degree of modularity, the energy router based on SST has received higher attention.

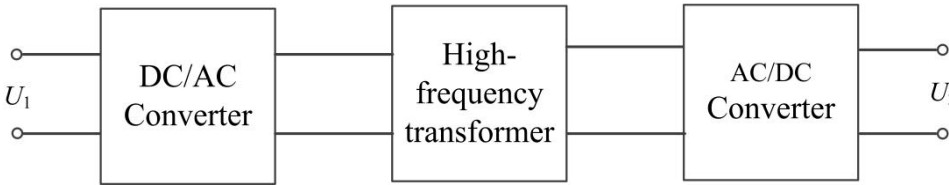

**Figure 2.** Basic architecture of energy router based on SST.

2. Multi-port energy router

The multi-port energy router connects the power grid, DG unit, energy storage unit and load unit through the DC or AC port provided by the high-efficiency power electronic equipment and realizes the access and consumption of renewable energy through the control of DG unit and new load, thus maximizing the comprehensive utilization benefits of distributed energy [21]. Therefore, developing multi-port energy routers meets the objective requirements in the new situation, as well as the strategic requirements of our country in terms of energy and the economy. Compared to the SST type energy router, this type is more suitable for small and medium power occasions where power conversion and distribution are needed in regional low voltage systems [22]. It often involves the transformation and combination of a variety of power electronic converters, as illustrated in Figure 3.

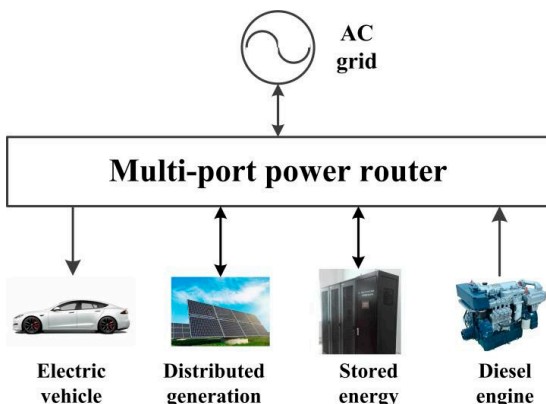

**Figure 3.** Basic architecture of multi-port energy router.

## 3. Energy router based on PLC

The energy router, based on PLC technology, utilizes traditional power line communication as the energy transmission carrier and operates in a time-sharing and multiplexing mode for different kinds of energy packets to transmit and distribute energy to various devices on the same power line, as shown in Figure 4. However, since the energy packet under this scheme is in intermittent transmission mode, filtering devices and energy storage facilities need to be added to the load side to ensure continuous power supply, which is suitable for home energy routers [22].

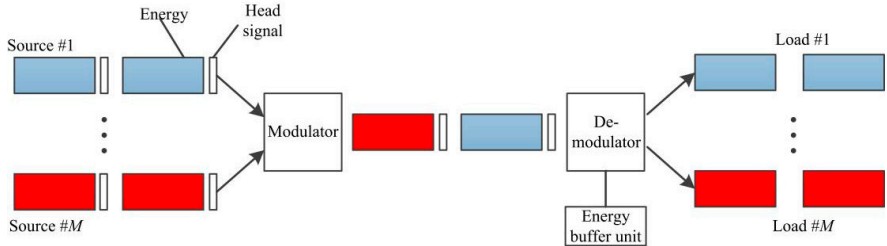

**Figure 4.** Energy router structure based on PLC.

The current research on energy routers at home and abroad mainly focuses on energy routers based on solid state transformers and multi-port energy routers, while the research on PLC in the field of energy routers is still in the exploratory stage, and there are few related research and applications. Because of its functional characteristics, multi-port power routers can well meet the needs of power users and make full use of renewable energy to realize economic benefits. Developing multi-port power routers is in line with the objective needs of the new situation and the strategic requirements of energy and the economy in our country, so it has practical application prospects and value. With the deep integration of cutting-edge digital information technologies such as distributed power generation technology, big data, and the Internet of Things, the requirements for friendliness, flexibility, intelligence, reliability, and stability of multi-port energy router applications are getting higher and higher [23]. On the basis of introducing the basic principle and typical structure of energy routers, this paper summarizes the research on key technologies of multi-port energy routers on topology type, analyzes the key problems in their development, so as to provide ideas for its follow-up research and development, and looks forward to their future development combined with the background of energy internet.

The following Figure 5 shows the basic classification of energy routers, the part inside the dotted line is the focus of this paper.

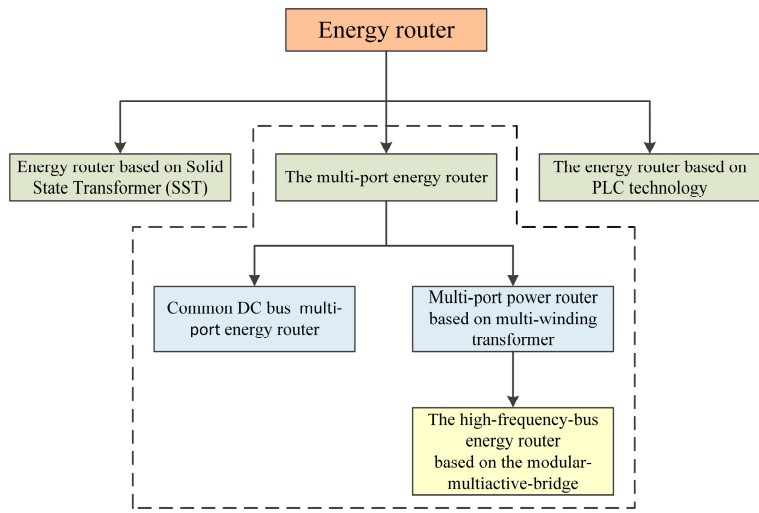

**Figure 5.** Basic classification of energy routers.

The rest of the paper is arranged as follows. Based on the topology structure, we sorted out the functional characteristics and related control technologies of various energy routers in Section 2. In Section 3, we introduce the development trend of energy routers, analyze the key problems in the development, and look forward to its future development. In Section 4, We have summarized the full text.

## 2. Typical Topology and Related Control Techniques of Multi-Port Energy Routers

Energy routers based on multi-port converters take electric energy as the core, form multi-port networks by connecting wind, light and other energy sources, and use multi-channel AC/DC, DC/DC and DC/AC power conversion to realize energy scheduling between "source-network-charge-storage" [24]. Therefore, multi-port energy routers are highly in line with the development needs of energy internet. From the perspective of the main circuit structure, the main circuit topology of the energy router is developed from the topology of SST, which is also the further evolution and application of SST in the future AC/DC power grid [25]. At present, energy routers based on multi-port converters can be divided into two categories according to the topology: (1) energy router based on common DC bus; (2) energy router based on multi-winding transformer [26–28].

### 2.1. Energy Router Based on Common DC Bus

Common DC bus type multi-port energy router is generally a combination of a number of independent AC/DC converters and DC/DC converters, its basic architecture is shown in Figure 6, high voltage and low voltage ports through the common DC bus to achieve power coupling, with strong flexibility and plasticity, only according to the different physical characteristics of the external design of the corresponding converter.

Literature [29] designed a single-bus multi-port energy router composed of several DC/AC converters and DC/DC converters. The topology is shown in Figure 7, which includes multiple ports such as photovoltaic, energy storage and wind power. It is simple in structure and easy to expand. In this scheme, energy flows between ports through common DC buses, but electrical isolation cannot be realized between different ports. Moreover, since energy storage and photovoltaic in the system are not always in a working state, the problem of multi-working condition operations will be involved. Hard switching of working conditions will cause a sudden voltage change of bus bars, but the scheme has not conducted research on working condition switching. Therefore, it does not meet the requirements of high reliability operation.

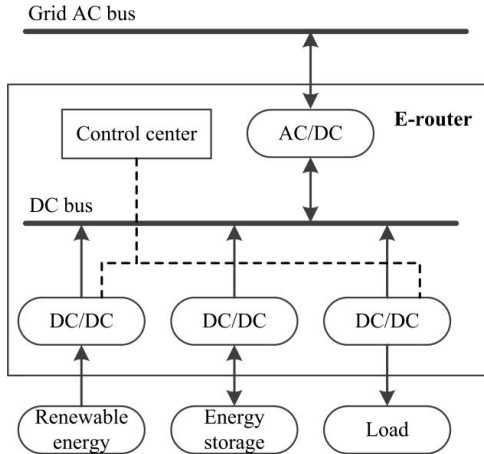

**Figure 6.** The Basic topology of energy router based on common DC bus.

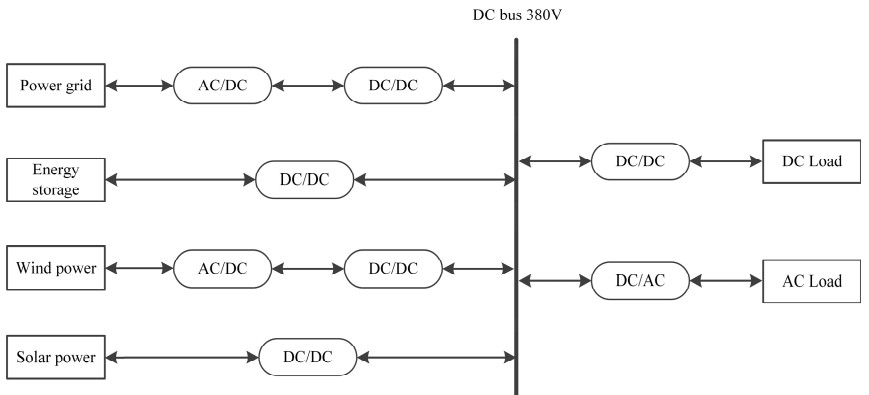

**Figure 7.** Energy router structure in the literature [29].

Aiming at the problem of energy router working condition switching during renewable energy access, the literature [30] divides the working mode of power router into four working modes: AC/DC power distribution, flexible interconnection, power flow transfer and self-stabilization, and analyzes the power flow under each mode, and proposes a coordinated control strategy based on virtual synchronous generator (VSG). Because the VSG control has both P/Q control and V/f control, it can be controlled with a full energy router and/or off-grid and isolated island operation without the need for mode switching. When the working condition is switched, it can avoid the adverse effect on power quality caused by the switching of the control mode. In this way, the flexible transfer and function of energy between multiple ports and converters at all levels in each mode are realized. However, each working mode in the scheme uses energy storage to control the DC bus voltage, and only when the energy storage battery capacity needs to be large enough, the bus voltage can maintain a small fluctuation during the working condition switching. Therefore, there are some limitations in practical application. The literature [31] innovatively introduced compressed air energy storage into the power router to improve the energy storage capacity of the system. Moreover, compressed air energy storage has the advantages of long working time, large energy storage capacity and environmental friendliness, which can well adapt to the characteristics of energy routers and future power systems. Therefore, this scheme can provide some research ideas for the development of multi-port energy routers.

The literature [32] carried out further research on the problem of power router working condition switching and port coordination and proposes a four-port power router scheme. Its main circuit is shown in Figure 8. The energy storage unit is connected to the 750 V DC bus by dual active bridge (DAB) converter, the DC load unit is connected to the DC bus by

DAB converter, and the photovoltaic unit is connected to the DC bus by a Boost converter. The grid-connected unit phase bridge voltage source converter (VSC) is connected to the DC bus to form a four-port energy router.

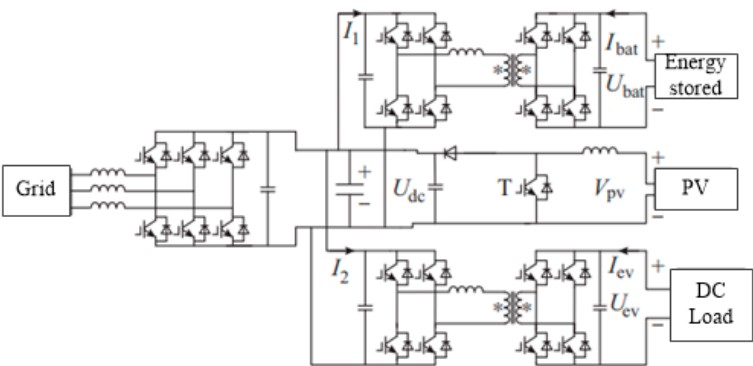

**Figure 8.** Energy router structure in the literature [32].

Based on the operating conditions, six operating modes are divided into grid-connected operation, grid dispatching and feeding, grid dispatching and absorbing electric energy, off-grid operation, battery protection and system shutdown. A unified and coordinated control strategy based on layers is proposed under the framework of energy internet. The control system is divided into three layers: power grid dispatching, microgrid control and local control, as shown in Figure 9, which are used to control the coordinated operation of the energy router under multiple operating conditions. A new droop control method based on current tracking is proposed, which realizes the accurate and controllable transmission power of energy storage, charging pile and DC bus, and indirectly realizes the controllable transmission power of VSC. The experimental results show that the seamless switching of the energy router under any working conditions and the stable control of the DC bus voltage are finally realized.

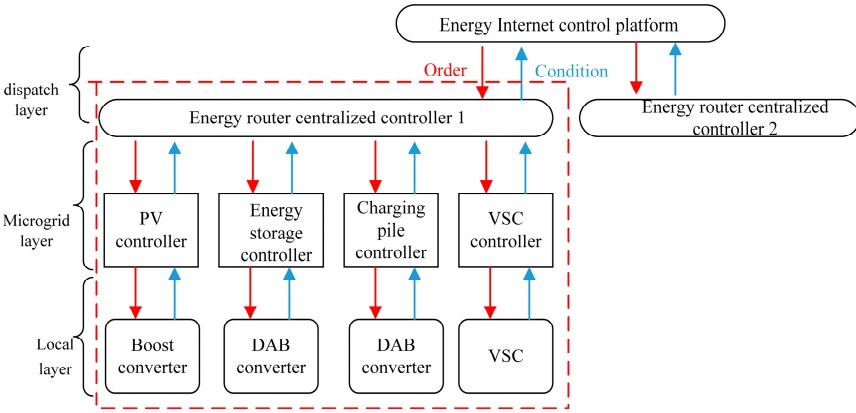

**Figure 9.** Hierarchical control framework in the literature [32].

In order to solve the disadvantages of the energy router with a single bus structure, which may have frequent bus voltage fluctuation and great impact on the power grid port and load port, the literature [33] proposed an electric energy router based on the dual DC bus architecture. Its topology is shown in Figure 10. In the form of a variety of ports, the photovoltaic, energy storage, AC power grid, AC and DC loads are interconnected. Dual bus architecture improves system reliability; however, there are many converters in this scheme, and some converters have a large voltage span when switching between different DC bus bars, so it is difficult to ensure the effective suppression of bus voltage fluctuation and the stable operation of each port.

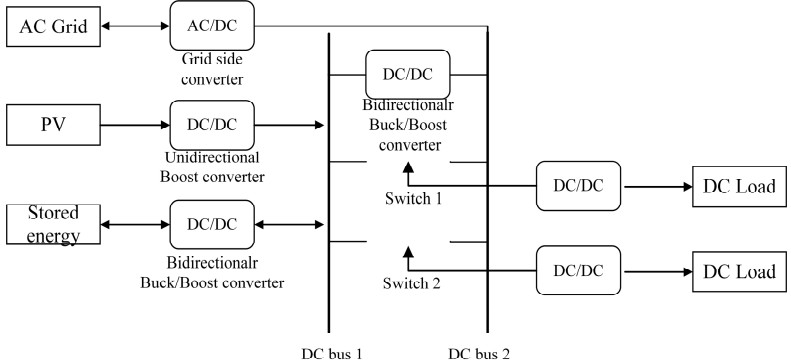

**Figure 10.** Energy router structure in the literature [33].

Based on the above topology, the literature [34] proposes a dual DC bus power router scheme, which introduces sliding mode control and active disturbance rejection control into the control of key components of the dual DC bus power router. Simulation results show that the bus voltage can still be maintained as stable, with high robustness and strong anti-interference ability under off-grid and grid-connected operating conditions. However, it is necessary to apply the control method to the hardware experiment platform to test the actual effect.

With the development of a power system, the system is developing towards AC-DC hybrid distribution with multi-DC bus and multi-power flow. The literature [35] proposes an AC-DC interconnection energy router scheme for medium and low voltage distribution networks. Its architecture is shown in Figure 11. It can provide multiple AC and DC busbars with different voltage levels and can provide plug and play ports for wind power, photovoltaic, energy storage, AC-DC load and microgrid. A mode switching strategy with lead compensation is proposed to realize seamless switching between grid-connected mode and island mode. However, there are many converters in this scheme, the hardware cost is high, the control system is more complex, and the port isolation problem is not considered. To achieve large-scale promotion and application, further research is needed to save costs, optimize control, and improve the overall stability of the system.

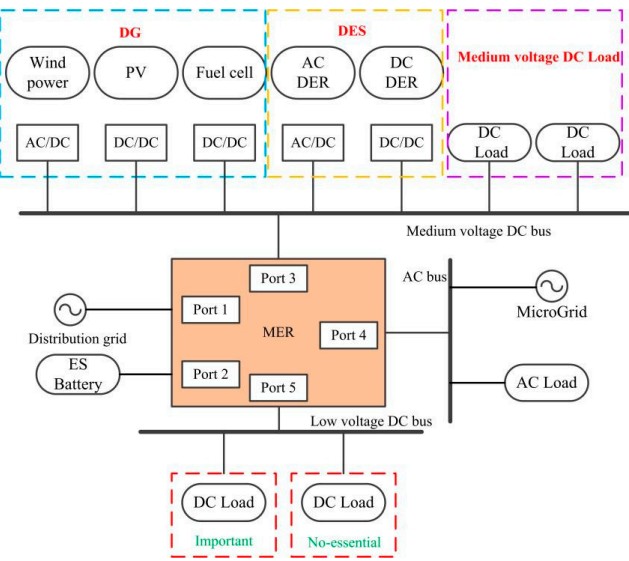

**Figure 11.** Energy router structure in the literature [28].

The common DC bus line energy router used in the medium and high voltage distribution network usually needs to adopt multilevel technology at the input end. At present,

there are two typical energy router topologies, namely, the cascaded H-bridge topologies (CHB) and the modular multilevel converter (MMC) topologies [13].

The typical structure of the energy router based on the CHB topology is shown in Figure 12. The high-voltage AC input adopts the cascade H-bridge rectifier circuit, and the isolation stage adopts a DAB converter. DAB has a wider voltage conversion ratio and higher power density, which can realize the bidirectional flow of the power flow and is conducive to energy transmission. Ac loads can also be accessed through the inverter. This structure can not only meet the requirements of high voltage and large capacity, but also provide DC ports, which meet the basic requirements of multiple ports required for power routing [36].

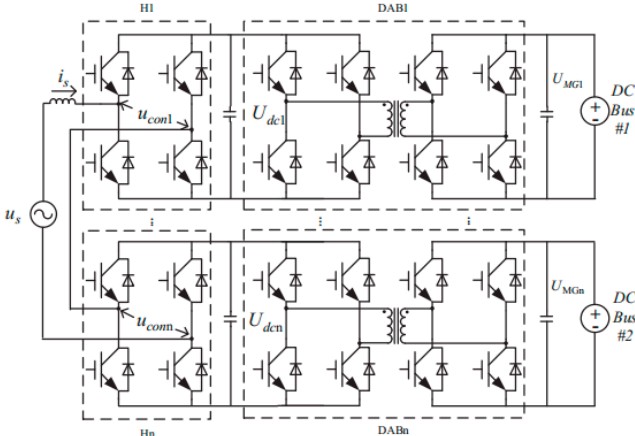

**Figure 12.** Energy router topology based on the CHB.

At the same voltage level and withstanding the voltage level of power semiconductor devices, the MMC-based energy routers require fewer switching devices and high-frequency transformers than the CHB topology and can have high-voltage DC ports to achieve richer access and interconnection functions, which is more suitable for future distribution network development needs [37].

The literature [38] proposes an energy router scheme based on MMC. Its architecture is shown in Figure 13. The power unit consists of multi-level power electronic conversion units, and the DC bus obtained through intermediate transformation can lead to multiple types of interfaces to facilitate the integrated access of multiple energy sources. The grid-connected port is connected to the power grid by MMC, the centralized photovoltaic is connected to the medium voltage DC bus through DC boost, and the isolation level uses DAB to connect the medium and low voltage DC bus through the mode of input series and output parallel. In order to solve the problem of medium and low voltage renewable energy consumption in island mode, The low-voltage DC bus is incorporated into the hybrid energy storage system (HESS) unit.

In terms of control, two operating modes, grid-connected and isolated island, are divided according to operating conditions. Aiming at mode switching of energy router and transient shock in the switching process, the switching time sequence of operation mode switching of energy router is designed, and a coordinated control strategy of mode switching based on the output compensation of the controller based on the shift ratio maintainer and current maintainer is proposed. By compensating the output of the bus regulator unit controller in advance, the adverse effects of control mode switching can be effectively reduced, and the transient stability of the system can be improved.

The literature [39] proposes an energy router scheme suitable for the interconnection of AC and DC power grids of multi-voltage levels. The main circuit topology is shown in Figure 14. The input stage is connected to the medium and high voltage AC power grid using MMC, the isolation stage is composed of two parts, and some DAB modules are connected to the AC ports of the output stage using ISOS structure. The other parts

are connected to the output DC port in ISOP structure. Compared with the isolator only using ISOP structure in the literature [38], the output stage can more easily meet the access requirements of AC-DC power grids with different voltage levels. The AC output port is connected to the 10 kV medium-voltage AC power grid with the MMC structure, and the DC port is connected to the low-voltage DC power grid with the Buck-Boost DC/DC converter. The AC/DC port of the output stage can be expanded according to the needs.

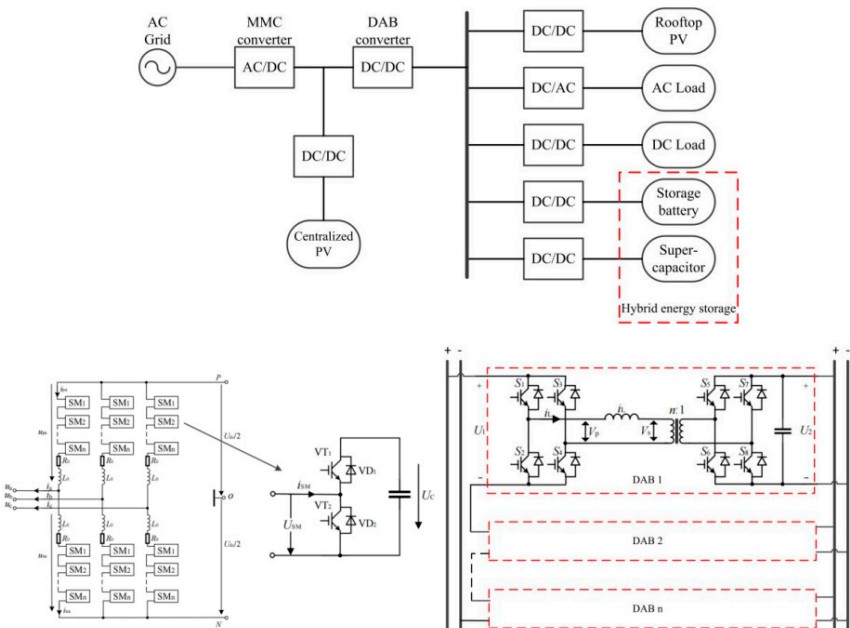

**Figure 13.** Energy router structure in the literature [31].

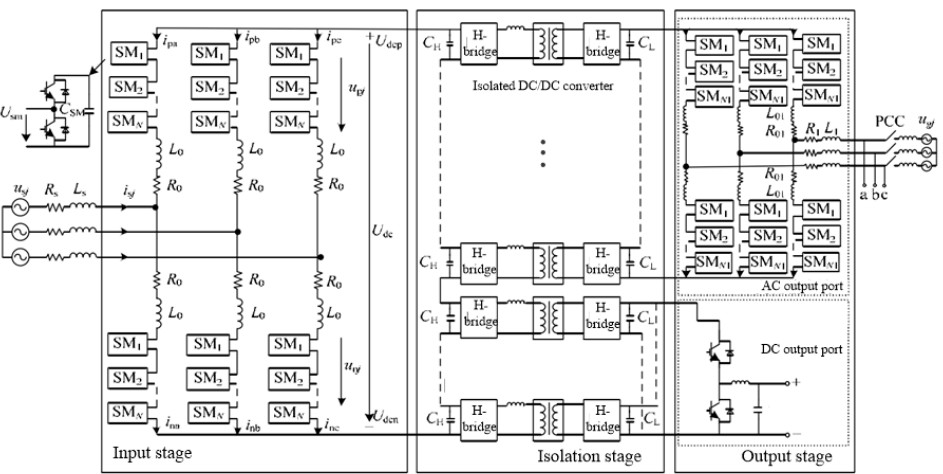

**Figure 14.** Energy router structure in the literature [39].

In terms of control, an input level control strategy based on virtual synchronous motor is proposed to increase the inertia and damping of the power grid and make the energy router more friendly to the upper power grid. A power coordination control method is proposed to adjust the DC voltage of the high voltage side of the input stage and the isolation stage to coordinate the power flow between the ports, to ensure the stable operation of the energy router.

Although the above power router scheme based on CHB and MMC has been widely used, there are still some problems to be solved. First of all, both of them do not solve the problem of electrical isolation of the port; secondly, for the multi-port energy router with cascaded H-bridge structure as the input stage, the AC voltage support at the network

side is provided by cascaded H-bridge converters, resulting in a large number of cascaded modules. In order to provide DC ports, DAB converters need to be added, which makes the system bulky and costly. Finally, although the multi-port energy router using MMC in the input stage can reduce the number of DC/DC high-frequency transformers in the isolation stage, the large number of MMC sub-modules on the AC side also brings the problem of large system size and high cost, and MMC can only realize the isolation between the high voltage port and the low voltage port. It is also necessary to design a complex control algorithm to solve the problem of submodule voltage balancing and circulation, and the control system is more complicated [40]. Table 1 shows how the two compare in terms of performance, reliability and economy.

**Table 1.** The comparison between CHB and MMC topology.

| Item\Topology | CHB | MMC |
|---|---|---|
| property | The output efficiency is slightly lower, and the control strategy to compensate the unbalanced current is more complex | High output efficiency, can provide medium and high voltage DC ports, there are circulation, capacitor voltage imbalance and other problems |
| reliability | higher | The number of submodules is large, so the reliability is low |
| economy | High frequency transformer is more in number, large in volume and high in cost | The number of switching devices is larger and the cost is slightly lower |

### 2.2. Multi-Port Energy Router Based on Multi-Winding Transformer

The energy router based on multi-winding transformer coupling is formed by connecting corresponding half-bridge or full-bridge units through the natural multi-port structure of the multi-winding transformer. Since the multi-winding transformer used is generally a medium-/high-frequency transformer, the voltage gain range and isolation capability of the device are better, and it can solve the electrical isolation problem of the port which common DC bus topology cannot overcome.

The literature [41] proposed a three-port DC energy router applied in the field of microgrid, as shown in Figure 15. The three-port DC energy router device is composed of a medium-frequency three-winding transformer and three groups of full-bridge units. Each full bridge unit provides a DC port for connecting DC microgrid busbars of different voltage levels. The power flow of the three ports is achieved by adjusting the phase deviation of the high-frequency square wave output to the winding by the full-bridge units and realizing that the power coordination control among different microgrids and the energy management is at the system level. However, because the ports in this scheme share the same transformer core, there is a coupling of transmitted power between the ports, which is not conducive to the expansion of the ports, but also increases the difficulty of the design of the control loop, and deteriorates the power quality of the ports.

Aiming at the problem of poor power quality of multi-winding transformer-type energy router, the literature [42] adds a power quality management port on the basis of its general structure, as shown in Figure 16, which is composed of a series-parallel hybrid power quality management structure. The series management module uses the PI controller to follow the rating of the voltage, and the parallel management module uses the PI controller and repeated control to track the rating of the current. On the one hand, the power quality management port ensures that the grid current is a symmetrical sine wave and corrects the power factor, that is, inhibits current harmonics, and compensates reactive power and asymmetric components; on the other hand, the low voltage AC side is maintained as a three-phase symmetric sine wave, that is, to reduce the impact of distributed voltage fluctuations and harmonics on the low voltage AC side power supply. Compared with the traditional power quality management scheme which only deals with voltage fluctuation and harmonics, this scheme has more abundant functions and has

certain application value for the construction of multi-energy complementary distribution networks in the future.

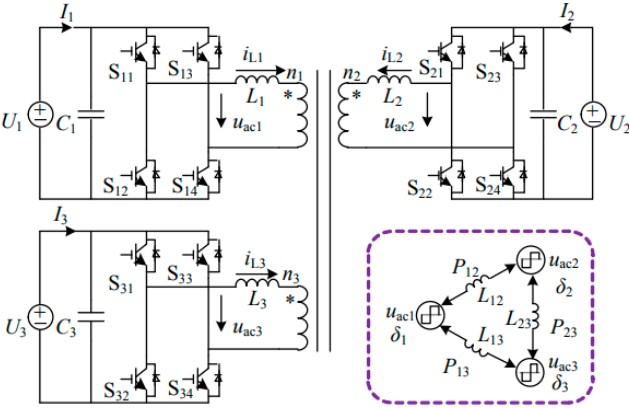

**Figure 15.** Energy router structure in the literature [41].

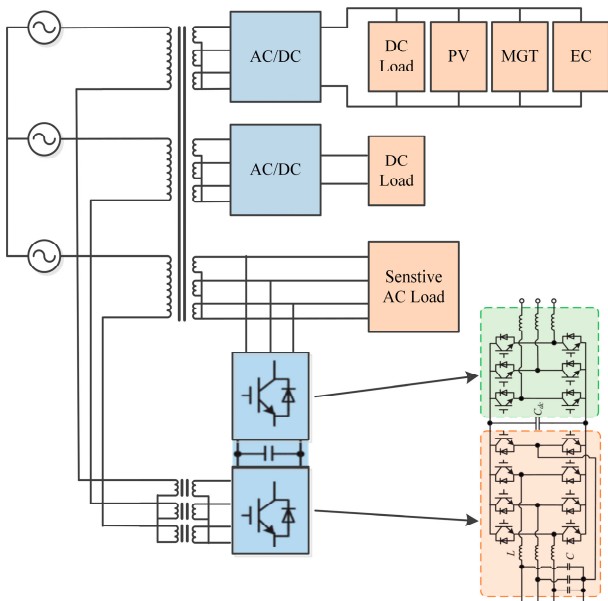

**Figure 16.** Energy router structure in the literature [42].

The literature [43] proposes a DC energy router based on cascade high-frequency transformers for small and medium-sized power situations with fewer ports, such as homes, commercial buildings and residential areas. Its topology is shown in Figure 17. Multiple high-frequency transformers cascade with each other, and H-bridge converters are merged into both ends of each high-frequency transformer, and the DC side of the converter is used as the DC port. The high-frequency transformer between the two adjacent ports can be thought of as a dual active bridge (DAB) converter, enabling soft switching characteristics and bidirectional energy flow. By reducing the number of converters and power conversion series, the topology reduces the cost and improves the transmission efficiency, and the port can also be expanded according to the actual demand. However, this scheme is only suitable for DC transmission, and each port has only one H-bridge, which makes it difficult to realize the power conversion of the high voltage level.

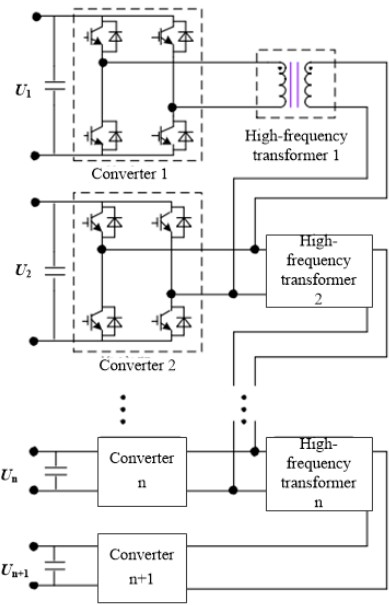

**Figure 17.** Energy router structure in the literature [43].

The topology of multi-port energy router based on multi-winding transformer has a complex coupling relationship between ports, so the manufacturing of a high-frequency transformer and design of a control strategy is difficult and the realization process is complicated. It is also difficult to carry out modular design and port expansion. At present, there are relatively few studies on this program, and the follow-up needs to improve the power and efficiency of energy transmission through advanced technical means on the basis of realizing basic functions.

In view of the aforementioned issues with the topology of the multi-winding transformer energy router, the literature [26,44] proposed an energy router topology based on the high-frequency convertor bus. The main circuit of this design is shown in Figure 18. The topology adopts a modular design, and its core is a modular multi active bridge (MMAB), each module comprises a high-frequency transformer and an H-bridge, forming the MMAB submodule. The AC sides of these sub-modules are connected in parallel by common low voltage and high frequency busbars, while the DC side can be extended into ports of any voltage/power level through a combination of series and parallel connections. In addition, any number of ports can be extended based on the common low-voltage high-frequency bus and ensure that the ports are isolated from each other. The structure represents an improvement of multi-winding transformer topology, offering significant advantages in modularity, expansibility, isolation and independence of each port. However, it also introduces challenges in circuit design, control and safety protection due to the difficulty in decoupling multiple stray parameters.

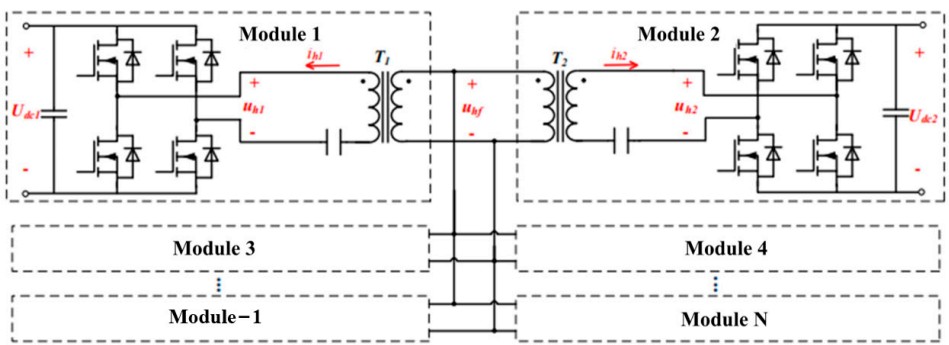

**Figure 18.** Energy router structure in the literature [26,44].

In view of the challenges in system control and protection of common high frequency bus topology, relevant studies have been conducted. As the foundation or crucial component of the operation of a common high-frequency bus energy router, its startup strategy must be able to overcome the complexity of the topology and meet the requirements of various application scenarios. Additionally, it is necessary to avoid problems that affect the safe operation of the equipment due to the inrush current and overvoltage. To solve this problem, the Literature [45] created a series of step start processes. These processes were designed by establishing the start priority and selecting the "main power supply" port under different operating conditions. Notably, the main circuit topology of the common high-frequency bus energy router remained unchanged throughout this process. Furthermore, a pre-charging strategy was designed based on gradually increasing the duty cycle and gave a mathematical description of the initial duty cycle and duty cycle increment. It provides theoretical support for the implementation of precharge control.

During the operation of the energy router, when the output voltage of the H-bridge arm changes abruptly, the integrated phase-shifting inductors and distributed capacitors inside the high-frequency transformer will generate free oscillation, resulting in a super-transient transition process of several MHz of high-frequency bus voltage, which is very easy to cause electromagnetic interference and trigger driver board failure. It has a serious negative impact on the reliability of high-voltage and high-capacity energy routers with modular scale. By analyzing the mechanism of the super-transient behavior, the literature [46] proposed a suppression measure of "uncontrolled rectification + damping resistance" in parallel on the high-frequency bus, and introduced the design method of damping resistance. The correctness of the theoretical analysis and the effectiveness of the suppression measure were verified by the test platform and the 10 kV/2 MW energy router industrial prototype.

In practical applications, the local module of the common high frequency bus energy router needs to block the device pulse outage due to fault protection or redundant control. However, despite this blocking, the module can still transmit power under the influence of high frequency oscillation caused by stray parameters, which may cause system security problems. In order to ensure the stable operation of the device, the literature [47] analyzed the system oscillation characteristics and module power transmission behavior after blocking the local module. This analysis was conducted by establishing the high-frequency oscillation model of the blocked module under the high-frequency bus structure. The study proposed measures to install the isolation switch on the AC side of the module and analyzed the influence of the stray parameters of the isolation switch on the oscillation characteristics and power transmission behavior of the blocked module. The accuracy of the model and the feasibility of the isolation switch are verified by simulation and experiment.

The above literature mainly focuses on the problems of system operation protection of the common high-frequency bus power router. At present, the decoupling between ports of the router is only applicable to situations where the number of ports is small, and the single phase-shift modulation strategy is adopted. However, there is little research on its power flow and energy management, which is not enough. And the above research basically stays in the stage of experimental prototype, and rarely in practical application in engineering.

Table 2 shows the comparison of the two main types of topologies for multi-port power routers. On the whole, compared with the multi-port energy router based on the multi-winding transformer and the common DC bus multi-port energy router, the former has better module expansibility and port independence at the same time, the number of power conversion links and components are relatively small, with advantages such as low cost and high-power density, but the latter has DC links inside. The coupling degree between the port converters of the system is lower than the coupling degree of the port converters based on the high-frequency bus topology, the overall inertia of the system is larger, the transient control is relatively easy, and the technology is relatively mature, so it is widely used [48]. The energy router is based on the topology of the multi-winding transformer, and it is also necessary to overcome the key technical problems, improve the

realizability of the scheme, reduce the difficulties of circuit design, system protection and control, so as to better apply to the field of multi-port energy router.

**Table 2.** Comparison of two types of topology of multi-port energy routers.

| Item\Topology | Common DC Bus Topology | Topology Based on Multi-Winding Transformer |
|---|---|---|
| advantage | The port can be expanded according to the demand, and can realize the access of various voltage levels, various forms of power supplies and electrical equipment, and the power level is higher | The electric energy conversion link is less, the port electrical isolation can be realized, the integration degree is higher, the power density is higher |
| shortcoming | The integration is low, the space is wasted, the electric energy conversion link is more, the coordination and control is difficult, and the efficiency is low | The coupling degree of the port is high, the decoupling is difficult, the hardware circuit design is difficult, and it is difficult to carry out modular design and port expansion |
| economy | The number of converters is large and the cost is high | The number of high-frequency transformers is small and the cost is low |
| applicability | It is suitable for low-voltage and high-power occasions, and has a wide range of application | It is mostly in the stage of laboratory prototype, with few application cases |

## 3. Development Trend and Prospect of Multi-Port Energy Router

The main circuit topology of the multi-port energy router is actually the development and evolution of the power electronic transformer topology, and as early as the 1960s, power electronic transformers based on power electronics technology have been studied by relevant scholars. However, due to the limitation of the development level of power semiconductor devices, the early development of power electronic transformers was relatively slow [49]. In recent years, thanks to the rapid development of power electronics and related technologies, the development of power electronic transformers has tended to be diversified. With the introduction of the concept of energy internet, the diversification and flexibility of electric energy conversion and transmission are further improved. It is necessary to realize the active selection of a power transmission path and the active control of a power flow direction in the power grid while implementing a controllable transformer, that is, the power routing function. In this context, power electronic transformers have evolved into energy routers: topologically, in order to adapt to the access of various power sources and loads such as DG, energy storage, and charging piles, the number of internal converters and ports has been expanded; in terms of control, by receiving the upper level scheduling instructions, the operation of the internal port converter is controlled, and the working data of each port is collected and uploaded. With the power routing function, the power conversion, power flow control and information interaction are realized.

Energy internet is based on the utilization of electric energy as the core, upgrading the traditional energy utilization mode by fully combining renewable energy generation technology, internet information technology and blockchain and other emerging technologies, coupling the use of electricity, heat, gas and other different energy sources, and achieving an energy interconnection and sharing network with cascade efficient use of energy on the output side. It is an important way to reduce environmental pollution, promote the consumption of clean energy, improve the comprehensive energy efficiency level of the system, and achieve China's "double carbon" goal [50]. The core connotation of the energy internet is the large-scale utilization of distributed renewable energy. "On-site consumption" is an effective way to utilize distributed renewable energy. As the basic core equipment of the energy internet, the energy router is a small power distribution system integrating distributed energy, energy storage, electric vehicles and energy conversion devices. It is also a feasible way to achieve friendly access and efficient use of distributed energy. The construction of energy internet needs to standardize the functions and roles of energy routers to provide technical support for access of a high proportion of renewable energy

and energy storage, advanced energy management and information exchange, bottom-up autonomous networking and user-oriented market model [51].

At this stage, the research on the energy router is not fully mature. As an important facility supporting the construction of energy internet, the future development of the energy router should start from its own functional characteristics, focusing on power conversion technology, control strategy and information communication technology to make breakthroughs and innovations, and achieve cross-field integrated application by combining with emerging technologies [12,52–54].

1. In terms of power conversion technology:

Power electronic technology is the core of the power conversion layer of the energy router. By designing the parameters of the electrical components and the combination of electrical unit modules in the internal power electronic converter, the voltage level transformation and electrical isolation can be realized. When the energy router is directly applied to the medium voltage distribution network, due to the limited pressure capacity of the existing power devices, on the one hand, high-power semiconductor devices based on new wide band gap materials such as SiC and GaN can be used to improve the voltage and power level of the power electronic conversion device, reduce losses, and save costs. On the other hand, it is necessary to combine the specific application scenarios, consider the pressure flow capacity of the energy router, electrical isolation, efficient access to renewable energy, AC-DC hybrid microgrid interconnection and other issues, and carry out the combined design of the converter to meet the overall performance requirements of the energy router.

2. In terms of control strategy:

Advanced control technology is the main means to realize controllable power conversion and active selection of power flow in the energy router. On the one hand, as a combination of various power converters, the energy router needs to adopt various control strategies to ensure the normal operation of each port converter and maintain the smooth operation of the whole system. On the other hand, it is necessary to combine the specific operating conditions, considering the characteristics of distributed power supply, power supply and demand balance, time-of-use price and other factors, by switching the working mode of the energy router in real time, the energy management of DG, power grid, AC and DC load, etc., to achieve flexible energy scheduling, and keep the power and voltage of each port stable during the input and output. However, as the structure of the power system becomes more and more complex, the uncertainty of distributed energy output and the randomness of the user's power consumption increases the challenge to the optimal scheduling of the energy router, and traditional methods cannot realize the accurate regulation of the energy router gradually. With the continuous upgrading of the performance of digital signal processors, a variety of non-traditional control strategies, such as neural network control, fuzzy control, predictive control, and other control technologies, can be applied to the complex control strategies of power grid transient processes, making the system control more flexible and diverse.

3. In terms of information communication technology:

In order to realize the accurate scheduling and management of electric energy, the energy router not only needs advanced power conversion and control technology, but also needs sufficient information and communication technology. At this stage, the power router still takes action according to the instructions of the superior dispatch center and lacks the ability to make autonomous decisions in emergency situations. In order to realize the active management and regulation of energy flow, it is necessary to integrate information technology, integrate communication capabilities into energy routers, network nodes and terminals at all levels, and realize intelligent power regulation. The 5th generation mobile communication technology (5G) has appeared, which can meet the technical requirements of multi-terminal, mobile, large-scale, high reliability, and low delay application scenarios,

and its application scenarios are highly coincident with the application scenarios of energy routers. The future energy router will combine 5G, artificial intelligence and other technologies to achieve real-time monitoring of energy and equipment on the power router, so that it has a better autonomous judgment function, to achieve the optimal decision in different situations, and improve efficiency, and be conducive to the efficient interaction between the user and the energy and equipment, enhance the user's participation in the energy internet.

4.   Cross-field integration of applications

With the development of various emerging technologies, there have been many collisions between the concept of the power router and other related technologies. Therefore, the application scenarios of the power router are constantly expanding, such as the power transformation and distribution system of urban rail transit, the emerging electric vehicle Vehicle to Grid (V2G) technology, etc. At the same time, this also provides an opportunity for the update of technologies in other fields, and then promotes the common progress of the whole engineering technology field.

- In the field of urban rail transit, as China puts forward the concepts of intelligent transportation and green transportation, the traditional rail transit power distribution system can no longer meet the development concept of the new rail transit. Therefore, energy feedback devices are added to the traditional rail transit power distribution system to reverse the energy during train braking back to the power grid. This option is technically feasible, but expensive. The application of energy router in the rail transit power supply system can, on the one hand, effectively solve the energy recovery problem in the braking process of rail transit by means of its multi-direction power flow function, without the need to add additional energy recovery devices; on the other hand, it can reduce the complicated multistage power transformation links of the rail transit power distribution system and improve the energy conversion efficiency of the system.
- Electric Vehicle to Grid (V2G) technology refers to the technology of electric vehicles to power the grid, and its core idea is to use the additional energy storage of a large number of electric vehicles as a buffer for the grid and renewable energy. This collides with the concept of power router. The combination of energy router and V2G technology can effectively solve the problem of large fluctuations in renewable energy, reduce the supply and demand pressure of the grid, and create benefits for electric vehicle users, which has great practical value and significance.

**4. Conclusions**

This paper introduces the basic concepts and main functions of energy routers, reviews the development and research status of multi-port energy routers based on the typical topology of energy routers, compares the two mainstream topologies of multi-port energy routers, analyzes its functional characteristics and performance, and discusses the important role of multi-port energy routers in the construction of future energy internet and power transformation under the background of energy internet and key technologies such as system control and information communication. Multi-port energy router meets the needs of energy internet construction and conforms to the concept of sustainable development. In the future, with the development and maturity of relevant technologies, multi-port power router will be combined with artificial intelligence, 5G, V2G and other new technologies to cope with various complex emerging challenges, and promote the integration of multi-industry development, to serve the safe and efficient supply of human society energy, will play an important role in the world energy reform and development.

**Author Contributions:** Author contributions: conceptualization, J.Z. and J.W.; methodology, J.Z. and J.W.; software, J.W.; validation, J.Z. and J.W.; formal analysis, J.W; investigation, J.Z. and J.W.; resources, J.W.; data curation, J.W.; writing—original draft preparation, J.W.; writing—review and editing, J.W.; visualization, J.W.; supervision, J.W.; project administration, J.Z.; funding acquisition, J.Z. All authors have read and agreed to the published version of the manuscript.

**Funding:** This paper was supported by the National Key R&D Program of China (No. 2021YFE0103800).

**Data Availability Statement:** Not applicable.

**Conflicts of Interest:** The authors declare no conflicts of interest.

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
