# Peer review of "Research Review on Multi-Port Energy Routers Adapted to Renewable Energy Access"

_electronics, doi:10.3390/electronics13081493_

Round 1

Reviewer 1 Report

Comments and Suggestions for Authors

Overall Assessment:

The manuscript presents a comprehensive overview of the development, current research status, and future trends of energy routers. It offers valuable insights into the typical topologies and control technologies of multi-port energy routers. The analysis is thoughtful, and the forecasting of key issues and future trends provides a solid foundation for subsequent research in this field. However, one area that requires significant improvement is the breadth and depth of the references cited.

Specific Comments:

Issue Identified: As a review article, this manuscript does not currently have a sufficiently broad or in-depth selection of references. This limitation limits the comprehensiveness of the literature review and may affect the credibility and scholarly contribution of the manuscript.

Recommendation: To enhance the manuscript's value and reliability, it is recommended to:

1.      Expand the Literature Review: Include a more extensive range of relevant literature. This should encompass seminal works in the field of energy routers, recent research articles that highlight the latest developments, and relevant theoretical and review articles that provide context and depth to the discussion.

2.      Diversify Sources: Consider incorporating references from a variety of sources, including but not limited to academic journals, conference proceedings, technical reports, and patents (if applicable). This diversification can enrich the discussion and present a more holistic view of the subject matter.

3.      Critical Analysis: Where possible, integrate references more critically. Rather than merely summarizing existing work, discuss how each reference contributes to the current understanding of energy routers, identifies gaps in the literature, or supports the manuscript's arguments and conclusions.

Author Response

I am very sorry that some unexpected errors have occurred. Please check the attachment submitted for the second time. The attachment submitted for the first time is invalid.

Reviewer 2 Report

Comments and Suggestions for Authors

The author reviews the technology of a Multi-Port energy router, introducing two main types: the Common DC Mode energy router and the Multi-Winding Transformer energy router. Concerning the Common DC Mode, it is noted that maintaining reliability in the face of rapid bus voltage fluctuations is difficult. As a solution, the paper describes research aimed at stabilizing DC voltages through four driving modes, the Four-Port Energy Router, Dual DC bus, AC-DC interconnection, as well as CHB (Cascaded H-bridge) and MMC (Modular Multilevel Converter) technologies. Additionally, research on the Multi Port energy router is presented, discussing the Three-Port DC, Cascade high-frequency transformer, and High-frequency converter bus, along with their respective drawbacks. The author has written a review paper, but upon reading it, I find it challenging to grasp an overall understanding of the technology. Upon reviewing the entire content, I perceive an overall lack of clarity in explaining the technology, and there are issues with the accessibility of some referenced parers. Furthermore, the strengths and weaknesses of each technology are not adequately delineated, making it difficult to grasp the key points of each. Below are the suggested revisions and recommendations after reading the manuscript.

[Comments]

1)       As this is a review paper, please ensure to reference papers that are universally accessible.

2)       Since there were some papers that couldn't be found, please consider revising the selection and titles of the referenced papers.

3)       The explanations regarding the technology are generally insufficient. For example, there is inadequate explanation regarding the techniques aimed at addressing the rapid fluctuations in bus voltage, despite mentioning the challenges of the Common DC Mode.

4)       There are frequent instances of grammatically awkward phrases or sentences that make it difficult to grasp the content. I recommend revising lengthy sentences into concise ones that encapsulate the main points, and overall, suggest editing for awkward English usage and grammar.

5)       Although the paper deals with the overall aspects of the technology, I recommend providing more detailed explanations of the strengths and weaknesses. When reading the paper, I feel there is insufficient clarity in understanding the strengths and weaknesses of the technologies discussed.

6)       When introducing specific technologies, I recommend assigning subtitles or numbers to clearly indicate the beginning and end of the content you wish to introduce.

7)       Lastly, apart from the points I mentioned, there are typographical errors, duplicate figure numbers, spacing errors, and other issues in the paper that require review.

-          1, 2 Ex) Difficult to approach [10],[24], [25], [26], [27], [30], [31], [33], [35], etc

-          3, 5 Ex) Explanation of electrical isolation cannot be achieved between different ports (Line 165), Provide the formal name of the first mentioned term(Line 183), Explanation of four operating modes(Line 171) etc (Explanation of [27], [28]…)

-          4 Ex) electrical physical layer (Line 59, 69), Explanation of the figure is needed in the text (Line 95), etc (175-176, 359, 354-388…)

-          6 Ex) Line 354-397, 170-184 etc

-          7 Ex) Figure 14 is three (Line 311, 325, 352), Center align the figure and caption, etc

Reviewer 3 Report

Comments and Suggestions for Authors

The abstract contains too general information. I recommend not to use the same keywords that are in the title. The structure of the article is not according to the template. Considering the presented structure, it is not clear where the introduction ends. In the beginning, I recommend consistent citation of references dealing with the issue of energy routers, indicating the specific focus of each publication (please state strictly the focus of the publication, presented result, authors, etc.). A comprehensive condensed summarization of the current state of the issue in the introduction will provide a better starting platform for comparison. In the introduction, I recommend clearly defining the goal of the article. In the methodology section, I recommend specifying the comparison criteria. discussion - please state specific advantages, disadvantages and comparisons (extension of table 1) In the article, it is necessary to formulate specific conclusions, benefits and recommendations for practice.

Round 2

Reviewer 2 Report

Comments and Suggestions for Authors

Thank you for your sincere responses to the previous questions. Most of the requested revisions have been made but there are still a few areas that require further attention. Below is a list of additional items that need to be reviewed based on your answers.

[Comments]

1)      It would be clearer for comprehension to structure each item in a subtitle format rather than listing numbers. (Line 95-132, Line 523-597)

Ex) Please organize subtitles in a format similar to the following example or in another suitable format:

[subtitle]

The energy router based on ---

[subtitle]

The multi-port energy router ---

2)      Line 579 is an example of point 4, so it seems necessary to succinctly include only the key content and features. Additionally, there should be a clear indication that Line 579 relates to point 4.

Ex) Please clearly indicate the specifics of point 4 in a format similar to the following example or in another suitable format:

[subtitle]

Engineering and technology:

- In the field ---

- Electric -------

3)      There are still occasional grammatical awkwardness and typos. Please search for and correct as many as possible. (Line. 218 voltage vource converter, Line 115)

Reviewer 3 Report

Comments and Suggestions for Authors

I confirm that the authors implemented all recommendations in the manuscript and agree to its publication.

Author Response

Thank you again for your review